# Atomic-scale age resolution of planetary events

L.F. White[1], J.R. Darling[1], D.E. Moser[2], D.A. Reinhard[3], T.J. Prosa[3], D. Bullen[1], D. Olson[3], D.J. Larson[3], D. Lawrence[3] & I. Martin[3]

Resolving the timing of crustal processes and meteorite impact events is central to understanding the formation, evolution and habitability of planetary bodies. However, identifying multi-stage events from complex planetary materials is highly challenging at the length scales of current isotopic techniques. Here we show that accurate U-Pb isotopic analysis of nanoscale domains of baddeleyite can be achieved by atom probe tomography. Within individual crystals of highly shocked baddeleyite from the Sudbury impact structure, three discrete nanostructural domains have been isolated yielding average $^{206}Pb/^{238}U$ ages of $2,436 \pm 94$ Ma (protolith crystallization) from homogenous-Fe domains, $1,852 \pm 45$ Ma (impact) from clustered-Fe domains and $1,412 \pm 56$ Ma (tectonic metamorphism) from planar and subgrain boundary structures. Baddeleyite is a common phase in terrestrial, Martian, Lunar and asteroidal materials, meaning this atomic-scale approach holds great potential in establishing a more accurate chronology of the formation and evolution of planetary crusts.

[1] School of Earth and Environmental Sciences, University of Portsmouth, Burnaby Building, Burnaby Road, Portsmouth PO1 3QL, UK. [2] Department of Earth Sciences, University of Western Ontario, London, Canada N6A 5B7. [3] CAMECA, Madison, Wisconsin 53711, USA. Correspondence and requests for materials should be addressed to L.F.W. (email: lee.white@port.ac.uk).

Isotopic heterogeneities within meteorite samples caused by shock metamorphism[1–3] complicate efforts to characterize and date these precious materials. This has resulted in conundrums regarding the timing of major planetary events, including the timing of lunar magma ocean crystallization[4], Martian volcanism[1,2] and impact bombardment of the inner Solar System[5,6]. Recently, the coupling of isotopic geochronology with microstructural analysis of robust accessory phases has greatly enhanced our ability to date highly shocked samples by crystallographically contextualizing micrometre-scale geochemical analyses. In zircon (ZrSiO$_4$), crystallographically deformed grains have been targeted in efforts to ascertain impact ages[7], whereas more pristine igneous grains are selected to establish timing of primary crystallization[1,2]. However, the occurrence of zircon within the Solar System is mostly restricted to high silica (granitoid) magmas, mafic differentiates, and highly metamorphosed meteorites[8]. Baddeleyite (monoclinic-ZrO$_2$) is much more widely occurring, reported from a wide range of terrestrial mafic and ultra-mafic rock types[9] and within Lunar[10,11], Martian[1], chondritic[12] and asteroidal (HED[13]) meteorites. Grains are often small (<20 μm), but develop a range of characteristic microstructures under shock-loading conditions which allow for contextualized dating of target grains[2]. Although challenging to separate from whole rock aggregates[14], recent advances of in-situ U-Pb isotope analysis have enabled the dating of this phase using volumes as small as 5 × 5 × 1 μm (secondary ion mass spectrometry[15]). However, given that shock microstructures in baddeleyite vary at the micrometre to sub-micrometre scale[1,2], such techniques still homogenize microstructural domains, yielding an array of partially reset ages within highly shocked populations[2]. Here we use atom probe tomography (APT) to accurately resolve chronological end-members (that is, protolith crystallization and impact metamorphism) within highly shocked baddeleyite of the Matachewan dyke swarm[16] (Ontario, Canada). This approach has the unique potential to produce coupled isotopic and structural data sets from nanoscale domains[17], facilitating direct U-Pb dating of nanometre-scale features.

## Results

**APT of unshocked baddeleyite.** Although atom-probe analysis of Pb isotope ratios ($^{207}$Pb/$^{206}$Pb) has previously been conducted on zircon[17], this study represents the first effort to analyse baddeleyite on the atomic scale. In this regard, reference ZrO$_2$ grains of the Phalaborwa carbonatite complex, South Africa[18] have been analysed under the same conditions as the unknown grain population. Spectral analysis identified the most abundant U-bearing compound as the species $^{238}$U$^{16}$O$_2^{2+}$ at 135 Da (mass-to-charge ratio in daltons), although smaller peaks were observed at 270 Da ($^{238}$U$^{16}$O$_2^+$), 84.66 Da ($^{238}$U$^{16}$O$^{3+}$) and 127 Da ($^{238}$U$^{16}$O$^{2+}$). Pb isotopes $^{206}$Pb, $^{207}$Pb and $^{208}$Pb in the 2+ charge state was ranged and quantified at 103, 103.5 and 104 Da respectively, although counts of $^{207}$Pb and $^{208}$Pb are below detection limits. This comprehensive analysis of U and Pb isotope systematics is facilitated by the absence of Si within the baddeleyite lattice, limiting a large portion of the isobaric interferences that prevent accurate U measurements in zircon[19]. Despite being a mineralogical U-Pb reference material[18], Phalaborwa baddeleyite displays micrometre-scale uranium variability, including low- and high-U domains and localized zonation[20]. Sampling a low-U domain yielded a below-average concentration of U and Pb for this specimen, minimizing counts and resulting in isotopic ratios with large analytical uncertainties. Despite this, the standard microtip yielded a $^{206}$Pb/$^{238}$U ratio of 0.48 ± 0.09, within uncertainty of small-volume laser ablation analysis[20]. All elements, including known minor and trace elements that are incompatible in the ZrO$_2$ lattice (such as Ti

and Fe) are homogeneously distributed throughout both annealed (400 °C for 1 h) and unannealed microtips (Supplementary Fig. 1). These observations provide a first-order insight into unshocked baddeleyite nanostructure, where heating alone does not encourage a redistribution of incompatible cations.

**APT of shocked baddeleyite.** The studied thin section (JD12SUD14) is from a highly shocked diabase dyke sampled ~500 m beneath the lower northern contact of the Sudbury impact melt sheet. The sample is uniquely well-characterized, with high precision thermal ionization mass spectrometry constraints on the timings of crystallization (2,473 ± 16 Ma (ref. 16)) and impact shock metamorphism (1,850 ± 1 Ma (ref. 21)), as well as petrologically constrained shock (>10 GPa (ref. 22)) and post-shock (>850 °C (ref. 23)) P–T conditions. Baddeleyite is abundant in the sample, with over 100 grains ranging from 15 to 330 μm$^2$ found in the exposed surface area (Supplementary Fig. 2). Field-emission scanning electron microscope (SEM) and electron back-scatter diffraction (EBSD) analyses of these grains reveal a range of microstructures, including complex mosaics of interlocking twin domains, partial amorphization and partial granularization (Fig. 1). These structures differ greatly from those observed in unshocked grains where simple and polysynthetic twinning is dominant. Two grains (44,755 and 46,059) were isolated for APT analyses. The two grains yielded a total of 11 statistically significant atom probe data sets comprising between 2.6 and 35 million total background-corrected ions. The specimens are chemically similar, comprising ~99% Zr and O (bulk compound ZrO$_2$) with a number of impurities in the form of Hf, Nb, Fe and Y. Other elements, including Ti, Si, Ta, P, Yb, U, Ca, Sc, Mn, Al, Mg and Pb are present in very minor quantities (Supplementary Data). $^{206}$Pb$^{2+}$ produces sharp, well-defined peaks totaling between 269 and 4,870 counts in all microtip specimens, and the decomposition of UO and UO$_2$ peaks yielded between 737 and 15,464 total counts of $^{238}$U.

$^{206}$Pb/$^{238}$U ratios from the bulk tips generate geologically reasonable ages between 2,435 (±1,075) Ma and 1,485 (±313) Ma (Supplementary Table 1), albeit with large internal counting statistics errors. Despite this, the bulk ages highlight a partially-reset population yielding ages between crystallization (2.45 Ga (ref. 16)) and impact (1.85 Ga (ref. 21)), although the youngest age (1,485 ± 313 Ma) fails to reflect either of these chronological end members. Closer inspection of the eleven APT three-dimensional reconstructions reveals three discrete nanostructural domain types, highlighted by variable distribution of incompatible elements (Si, Mg, Al, Yb), most clearly iron (Fig. 2): (1) homogenous-Fe domains in which all elements are homogeneously distributed throughout the regions, appearing nanostructurally identical to annealed and unannealed microtips of the Phalaborwa baddeleyite standard; (2) clustered-Fe domains in which incompatible cations form nanoclusters ~10 nm in size that can be clearly isolated by a 2% Fe isosurface and are evenly distributed throughout the domain; (3) planar and curvi-planar domains displaying ~10 nm nanoclusters enriched in all incompatible elements concentrated along their length. The latter features are either curvi-planar (interpreted as subgrain boundaries) or planar (probably associated with dislocation migration) and are often bound by incompatible-depleted regions within adjacent clustered-Fe domains. Five data sets comprise a single domain type (either homogenous-Fe or clustered-Fe), whereas the remaining data sets contain a mixture of the three defined subdomains (Supplementary Fig. 3). In these complex tips, nanostructural subdomains were first identified by Fe distribution before segregation using cuboid regions of interest, allowing separate mass-to-charge spectra to be generated for each region and yielding structurally isolated U-Pb ratios and ages.

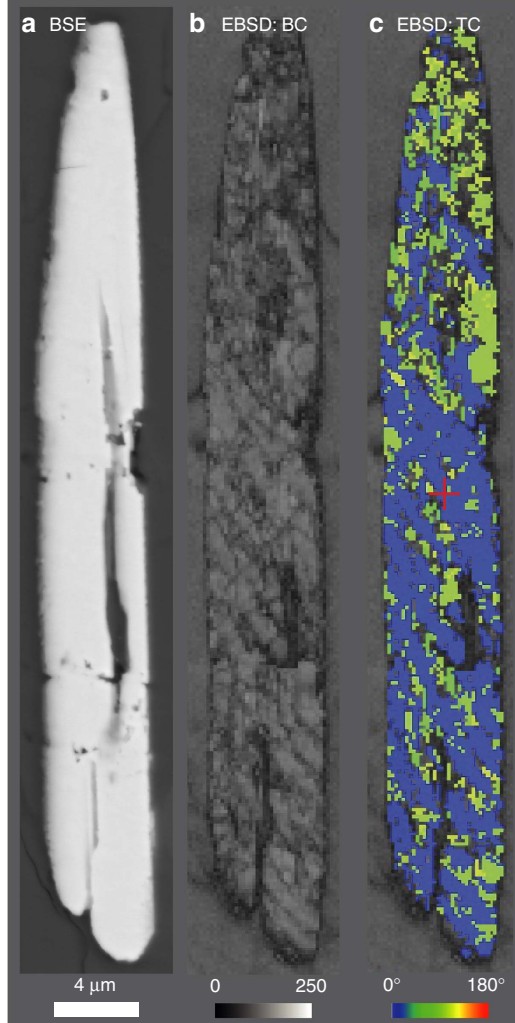

**Figure 1 | Microstructural data for Sudbury baddeleyite grain showing complex twinning and intensive crystal plastic deformation.**
(**a**) Backscatter electron (BSE) image of grain 8146, displaying a smooth appearance and absence of planar or curvi-planar features. EBSD maps (**b,c**) acquired at 200 nm step size: (**b**) band contrast (diffraction signal strength) map showing domains and planar features with weak diffraction owing to shock; (**c**) relative misorientation map of $m$-ZrO$_2$ component, highlighting $\sim$90° misorientation between the intensively crystal plastically deformed ($<$8°) master crystal and relatively undeformed subgrains. These less-deformed subgrains probably represent domains that have transitioned to the high-pressure, orthorhombic-structured polymorph of zirconia[26] and retrogressed to monoclinic-ZrO$_2$ during post-shock annealing.

This approach uniquely facilitates the generation of isotopic ratios from nanostructurally characterized subgrain domains and hence allows for evaluation of the variability in $^{206}$Pb/$^{238}$U ages within and between regions with differing nanostructural histories (Fig. 3; Supplementary Table 2). Homogenous-Fe regions (two full data sets and two subdomains) yield an average $^{206}$Pb/$^{238}$U age of 2,436 ± 94 Ma (3.9%; 2 s.e.), within uncertainty of the known age of the diabase protolith[16]. Isolation of the entire clustered-Fe domains (three full data sets and five subdomains) yield an average $^{206}$Pb/$^{238}$U age of 1,852 ± 45 Ma (2.4%), which is within uncertainty of the age for the Sudbury impact event[21]. Individual Fe-clusters within these domains yield low counts of U and Pb, which fall below the detection limits of APT. Isolation of four planar and curvi-planar features produces an average age of 1,412 ± 56 Ma (4%). This younger age overlaps with the timing of the last significant tectonic event to affect the region, the Chieflakian Orogeny, which resulted in widespread greenschist facies (*ca.* 350–500 °C, *ca.* 2–8 kbar) metamorphism across the study area[24]. Taking weighted averages of these same data sets yields similar ages and uncertainties, but with very low mean squared weighted deviation values (0.028–0.038) that suggest the internal counting statistics errors overestimate the true reproducibility of the measurements.

## Discussion

Partially impact age-reset grain populations have typically been inferred to represent variable grain to micrometre-scale diffusion of radiogenic isotopes during thermal metamorphism of shock-deformed domains[25], including for baddeleyite[8]. However, our findings show that the development of localized nanometre-scale structures controls the mobilization potential of Pb in baddeleyite and results in subdomains that preserve either protolith crystallization or impact-reset $^{206}$Pb/$^{238}$U ages. These nanostructures probably relate to partial formation of the high-pressure orthorhombic-ZrO$_2$ phase during shock loading of the crystal lattice[2,26]. This is known to be a progressive transition above $\sim$5 GPa (ref. 26), due to the anisotropic elastic properties of baddeleyite[27]. Transformed domains would readily revert to the stable monoclinic structure during melt-sheet induced annealing ($>$850 °C (ref. 23)) immediately following impact[28]. It is likely to be that nanoscale defects induced by these transformations facilitate the migration of Fe into clusters during heating, while also facilitating the complete diffusion of Pb from the domain. Preserved igneous $m$-ZrO$_2$ domains would not contain these pathways, preventing Pb and Fe diffusion within the lattice and preserving both the homogenous distribution of atoms and the primary U-Pb age. The planar and curvi-planar features observed appear more susceptible to Pb-diffusion during low temperature and pressure (tectonic) metamorphism, suggesting that these operate independently of the host baddeleyite grain. This observation is in agreement with previous studies, which suggest nanometre-scale planar and linear features undergo continued pipe diffusion of incompatible elements and isotopes below the closure temperature of the crystal[29]. An important implication of the observed nanostructural and isotopic complexity is that larger analytical volumes[15,20] will homogenize these domains and yield a mixed age from multiple isotopic reservoirs, which may not have clear geological significance.

Despite exposure to high-pressure shock metamorphism ($>$10 GPa (ref. 22)) and high-temperature post-shock annealing ($>$850 °C (ref. 23)), nanometre-scale domains in our sample preserve ages for both protolith crystallization (2,436 ± 45 Ma) and impact metamorphism (1,852 ± 145 Ma), suggesting that such data can be extracted from even the most highly shocked and annealed grain populations. The ability to generate targeted, high precision U-Pb ages with APT yields great promise when examining the tiny ($<$3 µm) baddeleyite grains which are reported within many planetary materials[1,2,10–12]. Going further, the ability to identify and sub-sample discreet nanostructural subdomains (for example, homogenous-Fe versus clustered-Fe) for isotopic analysis opens new avenues for dating highly deformed materials, including meteoritic samples. This allows for the unambiguous resolution of crystallization and impact ages in highly shocked planetary samples, presenting an exceptional opportunity to constrain timings for major Solar System events.

## Methods

**Grain location and imaging using SEM.** Baddeleyite grains were located *in situ* within a single thin section of JD12SUD14 using an automated feature scan of the

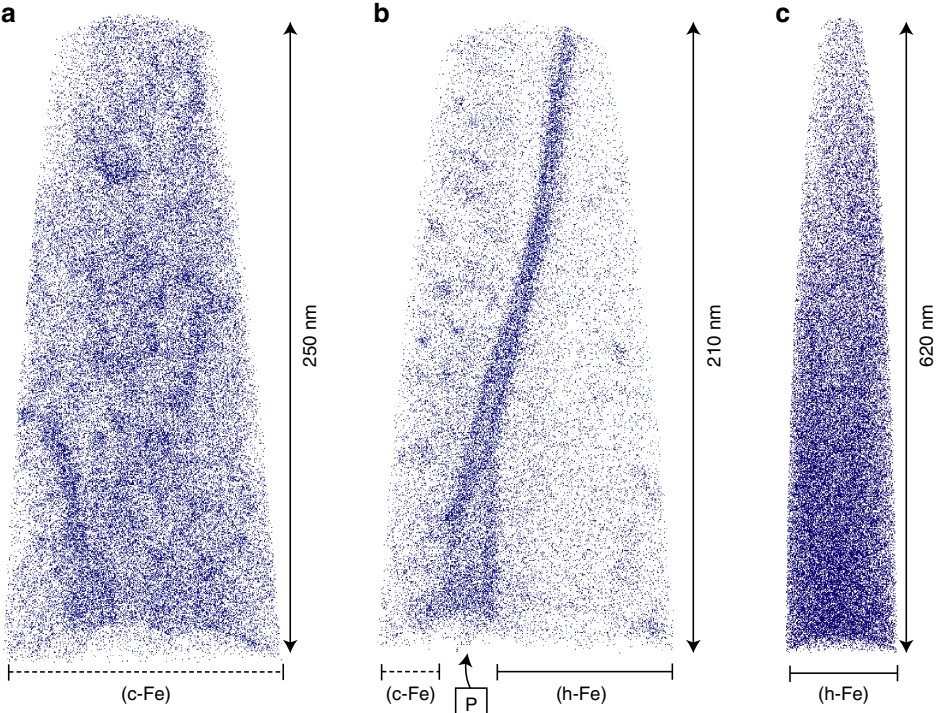

**Figure 2 | APT data for discrete nanostructural domains preserved within highly shocked baddeleyite sample JD12SUD14.** Three-dimensional APT reconstruction of three microtip specimens displaying distribution of individual Fe atoms. Three discrete nanostructural domain types are observed, one displaying homogenous distribution of Fe (h-Fe) and the other displaying nanoscale ($\sim$10 nm) clusters enriched in Fe (c-Fe). These domains are segregated by enriched planar and curvi-planar subgrain boundaries (P). (**a**) Microtip R60_146475 consists of a single clustered-Fe domain. The tip yields a bulk $^{206}$Pb/$^{238}$U ratio of 0.331 ± 0.115 in agreement with age data for the Sudbury impact event. (**b**) Microtip R60_146506 yields a bulk $^{206}$Pb/$^{238}$U ratio of 0.3493 ± 0.0741, although this purely represents a homogenization of multiple subdomains. Within this tip, the homogenous-Fe domain yields a $^{206}$Pb/$^{238}$U ratio of 0.4776 ± 0.3015, the clustered-Fe domain yields a ratio of 0.3525 ± 0.1154 and the enriched subgrain boundary yields a significantly lower ratio of 0.2555 ± 0.0712. A three-dimensional view of Fe distribution within this microtip can be seen in the Supplementary Movie file. (**c**) Microtip R60_144153 consists of a single Homogenous-Fe domain, representing the largest single tip within the dataset. The tip yields a $^{206}$Pb/$^{238}$U ratio of 0.4297 ± 0.1697 that is consistent with the age of dyke crystallization.

section coupling a Hitachi SU6600 field emission gun—SEM with an Oxford X-Max energy-dispersive spectroscopy detector housed within the Zircon and Accessory Phase Laboratory (ZAPlab) at the University of Western Ontario, Canada. EBSD analyses were conducted using an Oxford HKL Nordlys detector with step sizes of 150–250 nm (grain size dependant) closely following previously reported instrument conditions[8]. The only post-analysis noise reduction processing performed was to replace 'wild spikes' (interpreted as isolated, erroneously indexed pixels) with a zero solution.

**U-Pb analysis of baddeleyite using APT.** While EBSD analyses were ongoing two additional grains were isolated for APT. Microtip specimens were prepared using either an FEI Helios or Nova focused ion beam-SEM using standard lift-out and mount techniques to produce the desired specimen shape[30–32]. A series of annular mills with a decreasing inner radius were used to produce needle-shaped specimens with a radius of curvature of <100 nm. A total of 14 microtip specimens were created from four $\sim$2 × 10 μm pullouts derived from grains 44755 and 46059. In this work, the prepared specimens were analysed with a LEAP 5000 XR atom probe housed at the CAMECA Atom Probe Technology Center, Madison, Wisconsin. During acquisition, the specimen is placed under a high electric field, and evaporation and ionization of atoms from the surface occur commensurate with a laser pulse (125 kHz, 100 pJ). The mass-to-charge ratio of the ions is determined through time-of-flight mass spectrometry. Ion flight paths (382 mm) are terminated on a position-sensitive detector and the spatial information is reconstructed by projecting the ions back to the spherical end-form of the specimen and considering the sequential order of evaporation. Full details of local electrode atom probe data acquisition are provided elsewhere[23,33,34].

Of the 14 tips, 3 failed early (<100 nm run length), yielding statistically irrelevant data sets. The remaining 11 tips yielded between 2.3 × 10$^6$ and 3.5 × 10$^7$ background corrected, spectrally ranged ions over the length of the analysis. Tips ran smoothly, starting evaporation between 3 and 5 kV, and finishing between 8 and 12 kV. The exact beginning and end points depend primarily on the specimen size and shape after focused ion beam milling, and whether the experiment was

turned off or was stopped by fracturing of the specimen. Background counts are similar between tips, representing between 10% and 22% of the total (uncorrected) atomic counts. Thermal tails are present, but are generally only visible approximately one to three orders of magnitude down from peak height and so are only visible on the most prominent peaks, which have a large signal (Supplementary Fig. 4).

Nanostructural domains were defined visually based on the distribution of Fe within the reconstructed three-dimensional volume. Subdivision of domains yielded a total of 7.6 × 10$^7$ corrected, ranged ions within 8 Clustered-Fe domains, a total of 5.2 × 10$^7$ ions within four Homogenous-Fe domains and 2.5 × 10$^7$ ions within 4 planar and curvi-planar features. $^{206}$Pb$^{++}$ was measured at 103 Da. U-bearing compound peaks as 135 Da ($^{238}$U$^{16}$O$_2^{2+}$), 270 Da ($^{238}$U$^{16}$O$_2^+$), 84.66 Da ($^{238}$U$^{16}$O$^{3+}$) and 127 Da ($^{238}$U$^{16}$O$^{2+}$) were decomposed to yield total $^{238}$U counts for all data sets. For each individual peak, corrected ionic counts were calculated through the subtraction of background counts from the total (raw) ionic count. Peak location varied very subtly (<0.1 Da) between spectra and, as such, the exact mass/charge location of ranged peaks varies accordingly. In all scenarios, U and Pb peaks were ranged by eye from baseline to baseline, to ensure a reproducible count between microtip spectra. Propagation of absolute raw and background counting statistics errors ($\sqrt{\text{counts}}$) allows for estimated uncertainty on the background-corrected peak using equation (1).

$$\sigma \text{ Corrected counts} = \sqrt{(\sigma \text{Raw counts})^2 + (\sigma \text{Background counts})^2} \quad (1)$$

This yields the final error on $^{206}$Pb measurement, as it is derived from a single peak. As $^{238}$U represents the combination of UO and UO$_2$ peaks, a further step must be taken (equation (2)).

$$\sigma \text{U}^{238} = \sqrt{(\sigma \text{UO})^2 + (\sigma \text{UO}_2)^2} \quad (2)$$

Once individual error for $^{206}$Pb and $^{238}$U have been calculated, isotopic errors can be extrapolated in quadrature to yield an estimated error for the final $^{206}$Pb/

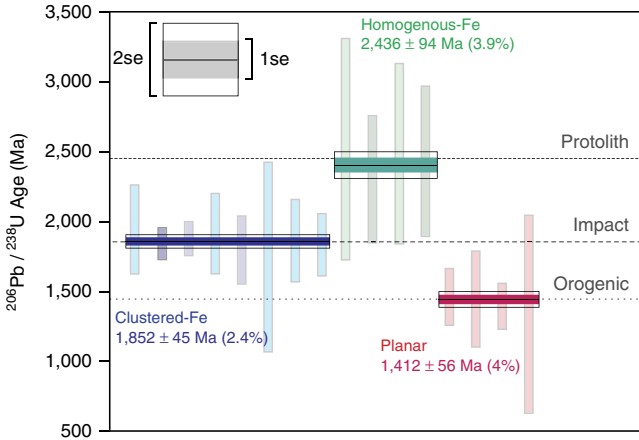

**Figure 3 | APT age data for all 11 microtip specimens subdivided by nanostructural domain, yielding accurate and precise ages for both crystallization and impact events.** These complex tips yield a series of partially to fully reset bulk U-Pb ratios, although separate subdomains are shown here to record both crystallization and impact ages. Among the 11 largest microtips, 8 domains display clustering of Fe (blue data) and yield an average $^{206}Pb/^{238}U$ age of 1,852 ± 45 Ma (2 s.e.m.), within error of impact age[21]. Four display homogenous Fe distribution (turquoise data) and yield an average age of 2,436 ± 94 Ma, overlapping with the known age for protolith crystallization[16], whereas four planar and curvi-planar subgrain boundaries (red data) record an average age of 1,412 ± 56 Ma, in agreement with the youngest orogenic activity in the area. Darker vertical bars represent bulk microtip analyses, whereas lighter data represent sub-divided structural domains within a mixed microtip specimen ($1\sigma$ counting statistic uncertainties).

$^{238}U$ ratio (and hence age) using equation (3).

$$\sigma\frac{Pb^{206}}{U^{238}} = \sqrt{\left(\left(\frac{\sigma Pb^{206}}{Pb^{206}}\right)^2 + \left(\frac{\sigma U^{238}}{U^{238}}\right)^2\right)} \times \frac{Pb^{206}}{U^{238}} \quad (3)$$

Average ages for all subdivided nanostructural domains were calculated by taking the mean $^{206}Pb/^{238}U$ age of each tip and/or characterized domain. Presented error bars for average ages (Fig. 3) represent 2 s.d. of the mean and include decay constant uncertainties. Mean standard working deviation values were calculated by taking a weighted average of the age population using isoplot[35].

**Data availability.** The authors declare that the data supporting the findings of this study are available within the paper (and its Supplementary Information files).

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

## Acknowledgements

We gratefully acknowledge the support of CAMECA in conducting APT analyses. This project was supported by Royal Society Research Grant RG160237, the Elspeth Matthews Fund of the Geological Society of London and a University of Portsmouth RDF Grant to J.R.D., and Canadian NSERC Discovery Grants to D.M. We thank I. Barker for his expert assistance with SEM analyses, Vale and Xstrata for assistance with fieldwork and two anonymous reviewers whose insights helped to improve this manuscript.

## Author contributions

All authors contributed to this work. D.E.M. and J.R.D. designed the initial project. All authors conducted portions of either, or both, the fundamental SEM and APT data collection and processing. L.F.W., D.A.R., D.E.M. and J.R.D. reduced and interpreted the APT data. L.F.W. reduced the SEM data. L.F.W. wrote the main paper and all authors discussed the results and commented on the manuscript at all stages.

## Additional information

**Competing interests:** The authors declare no competing financial interests.

**Publisher's note**: 

