## [Peer review File · Nature Communications]

Reviewers' Comments:

Reviewer #1 (Remarks to the Author)

Summary – This paper reports U-Pb dating of nano-scale domains in baddelyite grains that experienced shock during the Sudbury impact. The grains were analyzed using atom probe tomography, a fairly new analytical method. The study finds that the baddelyite grains contain three types of domains, classified by how Fe (and other elements) are distributed in 3D: 1) homogenous domains, 2) domains with clumped Fe, and 3) planar features. The study shows that the three domains each have an age that corresponds to a geologic event: 1) the crystallization of the grain in the homogenous domains, 2) the age of the impact in the clumped domains, and 3) a later orogenic deformation in the planar features.

General comments – The paper is an impressive display of the promise of APT. The correspondence of the ages of the domains with the crystallization of grains and the impact is very encouraging. Along with John Valley's paper on APT of the Jack Hill's zircon, this seems to be the opening salvo in a new era of geochemical analyses.

Specific Comments

1) In the clustered Fe domains, was all the U and Pb in the entire domain used to obtain the U-Pb age? Or just in the clusters themselves? From the text, it seems like it was in the entire domain. In which case, how does the domain record the age of the impact? To do that, there needs to be some change in the parent/daughter ratio in the domain. But if all that happened was that the U and Pb changed their 3D distribution, then U/Pb isotope ratios should not diverge from the initial material. Perhaps there was Pb loss from the clusters and/or matrix material within the clustered domain that changed the overall domain's U/Pb ratio? Whatever the case, this issue needs to be discussed in more detail. And if the clusters were not dated, seems like an obvious thing to do, if there are enough detected ions.

2) Along those lines, seems like the shock would be too fast for the Fe and other atoms to move into clusters. Even on the nano-scale, it requires diffusion of the atoms. Seems more likely that the shock altered the structure of the grain, and that over geologic time, Fe and other atoms diffused into the clusters, similar to Valley's model for the formation of the TE clusters he found in zircons, where alpha recoil damaged the structure of the crystal and then atoms diffused into it. Perhaps it is as atoms are diffusing into the damaged areas that the U/Pb ratio was changed. Given that the age of the domain is similar to the impact age, seems like this did not occur too long after the impact. The paper would be stronger if it discussed the mechanism of Fe clusters a bit more.

3) I think more detail about the mass/charge spectra is needed and there should be at least one spectra shown for each type of domain. Are there thermal tails on the peaks? Do they all have similar backgrounds? How were the boundaries of the ranges (low and high) chosen? Was it exactly the same for each needle? Until APT methods become more routine, I think the spectra and the ranging need to be shown. Wouldn't hurt to show voltage vs time curves. Do all the needles begin and end at the same voltage? And should make the reconstructed atom positions files available so other people can look at the data. Need to give enough data for people to reproduce results.

4) Would be interesting to see the clustering of other elements, or at least some indication of the concentration of the other elements in the domains and in the Fe-clusters. Perhaps that would give a clue to how the clusters formed.

Reviewer #2 (Remarks to the Author)

Review of "Atomic-Scale Age Resolution of Planetary Events" by White et al.

Impact events are considered to have played a significant role in the formation and evolution of planets, but precise and accurate dating of the events remains highly challenging. This is mainly because impact-related materials exhibit highly complicated textures and isotopic heterogeneities. The present study demonstrates the capabilities of atom probe tomography (APT) applied to baddeleyite U–Pb chronology for dating impact events. Analyzed baddeleyite grains were sampled from a highly shocked diabase dyke from the Sudbury impact structure where the ages of protolith crystallization, impact, and regional tectonic metamorphism are precisely known. The U– \rightarrow Pb and other trace element data revealed that three nano-structural domains can be identified in the baddeleyite grains based on the distribution of Fe and that the three discrete domains yielded distinct U–Pb ages corresponding to the ages of crystallization, impact, and regional metamorphism, respectively. The results indicate that nano-scale crystal structures control the migration potential of Pb and, therefore, that nm-scale analytical spatial resolution is critical for retrieving geologically meaningful ages from baddeleyite experienced multiple thermal events. Considering that baddeleyite is a common accessory mineral in a wide range of terrestrial and meteorite samples, the APT baddeleyite U–Pb dating has the great potential to be a powerful tool for chronology of planetary events. The data presented here are of high quality, the discussion hits all the important points, and the paper is very well written. I recommend the paper for publication in Nature Communications with minor revisions. I have only a few minor comments and suggestions.

l. 26–29: The present data record three, rather than two, discrete events (i.e., crystallization at 2436 ± 94 Ma, impact at 1852 ± 45 Ma and regional metamorphism at 1412 ± 56 Ma).

l. 44: Meteorite zircon occurs in highly metamorphosed rocks as well (Haba et al., 2014 EPSL; Iizuka et al., 2015 EPSL).

l. 67–69: Please state in the main text that signals of ^{207}Pb and ^{208}Pb are essentially under detection limits. Otherwise the reader would wonder why $^{207}\text{Pb}/^{206}\text{Pb}$ ages and concordant plot are not presented.

l. 78–79: Hafnium is compatible in the ZrO_2 lattice, as evidenced by a very high distribution coefficient (Klemme and Meyer, 2013 Chem. Geol.).

l. 86–88: It would be better to explain how the timings of crystallization and shock metamorphism were determined.

l. 88–90: I would suggest that a back-scattered electron image showing the occurrence of baddeleyite in the thin section is presented in the Supplementary Information.

l. 103: U238 should be ^{238}U .

l. 113–114: It would be beneficial to present a figure showing the distribution of Fe in the Phalaborwa baddeleyite standard (likewise Fig. 2) in Supplementary Data for comparison.

l. 317: Delete “d” in “(b,c,d)”.

Reviewer 1, Comment 1

1) In the clustered Fe domains, was all the U and Pb in the entire domain used to obtain the U-Pb age? Or just in the clusters themselves? From the text, it seems like it was in the entire domain. In which case, how does the domain record the age of the impact? To do that, there needs to be some change in the parent/daughter ratio in the domain. But if all that happened was that the U and Pb changed their 3D distribution, then U/Pb isotope ratios should not diverge from the initial material. Perhaps there was Pb loss from the clusters and/or matrix material within the clustered domain that changed the overall domain's U/Pb ratio? Whatever the case, this issue needs to be discussed in more detail. And if the clusters were not dated, seems like an obvious thing to do, if there are enough detected ions.

Reviewer 1 asks for further discussion regarding how the Clustered-Fe domains were dated, and highlights the potential implications for dating individual clusters rather than the whole (clustered) domain to the interpretation of our data. We thank the reviewer for the comment, and have taken the chance to strengthen our wording on subset selection and quantification within these Clustered-Fe domains (Line 132 - 133), further stressing that we determine U-Pb ratios from the entire Clustered-Fe domain and not individual clusters (which are unresolvable given that U and Pb counts are typically below detection limits in single clusters; Lines 135 - 136). As a result, we prove that Pb has been mobilized and lost from the domain, and not simply moved into clusters, explaining the impact-reset U-Pb age recorded by the domain.

Reviewer 1, Comment 2

2) Along those lines, seems like the shock would be too fast for the Fe and other atoms to move into clusters. Even on the nano-scale, it requires diffusion of the atoms. Seems more likely that the shock altered the structure of the grain, and that over geologic time, Fe and other atoms diffused into the clusters, similar to Valley's model for the formation of the TE clusters he found in zircons, where alpha recoil damaged the structure of the crystal and then atoms diffused into it. Perhaps it is as atoms are diffusing into the damaged areas that the U/Pb ratio was changed. Given that the age of the domain is similar to the impact age, seems like this did not occur too long after the impact. The paper would be stronger if it discussed the mechanism of Fe clusters a bit more.

Reviewer 1 suggests that Fe clustering may have occurred along nano-scale lattice defects induced by the impact event over geological time, similar to Pb clusters observed by Valley et al., (2014) in radiation damaged zircon. They continue by requesting further discussion on the

mechanisms of Fe clustering within the baddeleyite microtips. We thank the reviewer for the detailed comment and suggestion, and agree that the mechanism broadly involves impact deformation followed by diffusion. Within the Clustered-Fe domains, the transition from igneous monoclinic-ZrO₂ to orthorhombic-ZrO₂ during the impact event (and subsequent reversion to monoclinic at ambient conditions) likely generated an array of nano-scale diffusion pathways which allowed the Fe cations to migrate into clusters during post-impact heating. Homogenous-Fe domains that did not undergo this transition produce no such diffusion pathways, preserving igneous distribution of elements despite being exposed to the same severity and extent of annealing. As a result, we demonstrate that annealing alone is not sufficient to reset the U-Pb age of the domain, instead requiring impact induced nano-structures to facilitate cation diffusion and clustering. We have strengthened our discussion to better convey these points, and to this end Lines 151 - 161 reflect new text designed to avoid further confusion.

Reviewer 1, Comment 3

3) I think more detail about the mass/charge spectra is needed and there should be at least one spectra shown for each type of domain. Are there thermal tails on the peaks? Do they all have similar backgrounds? How were the boundaries of the ranges (low and high) chosen? Was it exactly the same for each needle? Until APT methods become more routine, I think the spectra and the ranging need to be shown. Wouldn't hurt to show voltage vs time curves. Do all the needles begin and end at the same voltage? And should make the reconstructed atom positions files available so other people can look at the data. Need to give enough data for people to reproduce results.

We agree with Reviewer 1 that further details on the acquisition and ranging of the mass to charge spectra would be useful, given the novelty of the technique. With this in mind, we've fleshed out the details regarding spectrum shape (i.e. thermal tails, background estimates), ranging, and peak selection provided in our Methodology (Lines 219 - 221 and 230 - 233). We've also included a fully labelled bulk mass/charge spectrum for a single microtip (#146506) along with smaller spectra for each subdomain (C-Fe, H-Fe, Planar) within this specimen (Supplementary Figure 4). Voltage versus time curves aren't normally shown for APT datasets (see Peterman et al., 2016; Piazzolo et al., 2016; Reddy et al., 2016), and we feel these would not strengthen the methodology presented here, though we've included further details within our expanded Methods sections (Lines 214 - 218).

Finally, to address the reviewers request for raw data to be made available. The position files (.pos) generated by APT contain a huge amount of data beyond the U-Pb data presented in this manuscript (full mass spectrum) and we are currently analyzing other isotopic systems within these datasets for future publication. As such, it is premature to release the full raw data files at this time. However, we feel that including a full data table of background corrected ions for all eleven tips (Supplementary Data), and a full break down of U-Pb counts and ratios (Supplementary Tables 1 & 2) is sufficient for a reader to judge the robustness of the data. If you feel it is absolutely mandatory to include raw data files, we would be willing to release one .pos file so that interested readers could recreate our subdomain selection as a proof of concept, though we would like to add that no other geological atom probe study has included raw .pos files with published manuscripts to date.

Reviewer 1, Comment 4

4) Would be interesting to see the clustering of other elements, or at least some indication of the concentration of the other elements in the domains and in the Fe-clusters. Perhaps that would give a clue to how the clusters formed.

Added a note on the composition of these clusters on Line 111.

Reviewer 2, Lines 26 - 29

The present data record three, rather than two, discrete events (i.e., crystallization at 2436 ± 94 Ma, impact at 1852 ± 45 Ma and regional metamorphism at 1412 ± 56 Ma).

We originally quoted two events (igneous crystallization and impact metamorphism) to highlight the application of our atom probe approach to dating 'planetary scale' events, though we take Reviewer 2's point that this somewhat undersells the strength of the technique. We now

reference all three discrete age domains (2436 ± 94 Ma, 1852 ± 45 Ma and 1412 ± 56 Ma) as the reviewer suggests (Lines 22 - 23).

Reviewer 2, Line 44

Meteorite zircon occurs in highly metamorphosed rocks as well (Haba et al., 2014 EPSL; Iizuka et al., 2015 EPSL).

We thank the reviewer for the comment, and have added the suggested comment and reference (Line 41).

Reviewer 2, Lines 67 - 69

Please state in the main text that signals of ^{207}Pb and ^{208}Pb are essentially under detection limits. Otherwise the reader would wonder why $^{207}\text{Pb}/^{206}\text{Pb}$ ages and concordant plot are not presented.

We take the reviewers point that without directly addressing the low concentrations of ^{207}Pb and ^{208}Pb the absence of concordia plots is confusing. We now address this directly within the paper (Lines 67 - 68).

Reviewer 2, Lines 78 - 79

Hafnium is compatible in the ZrO_2 lattice, as evidenced by a very high distribution coefficient (Klemme and Meyer, 2013 Chem. Geol.).

We no longer quote Hf as an incompatible element (with Fe and Ti) when discussing element distribution within the microtip specimen (Line 78).

Reviewer 2, Lines 86 - 88

It would be better to explain how the timings of crystallization and shock metamorphism were determined.

We now explain how high precision crystallization and impact U-Pb ages were obtained for the Matachewan dyke swarm, with clearer reference to the relevant geochronological papers and techniques (Lines 85 - 86).

Reviewer 2, Lines 88 - 90

I would suggest that a back-scattered electron image showing the occurrence of baddeleyite in the thin section is presented in the Supplementary Information.

We now show backscatter electron (BSE) images of grains #46509, #44755, #12133 and #62039 in the new Supplementary Figure 2.

Reviewer 2, Line 103

U238 should be ^{238}U . Corrected (Line 103).

Reviewer 2, Lines 113 - 114

It would be beneficial to present a figure showing the distribution of Fe in the Phalaborwa baddeleyite standard (likewise Fig. 2) in Supplementary Data for comparison.

We agree with Reviewer 2 that it would be useful to also show the distribution of Fe within the Phalaborwa baddeleyite standard microtip. We now show Fe and Zr distribution within a single tip of Phalaborwa to highlight the homogenous distribution of compatible and incompatible cations within an unshocked baddeleyite lattice (Supplementary Figure 1).

Reviewer 2, Line 317

Delete "d" in "(b,c,d)". Corrected (Line 389).

We trust that everything is in order, and that the changes and replies presented satisfy any concerns regarding the paper. Please do not hesitate to contact me if you need anything further.

Sincerely,
Lee Francis White (on behalf of all other co-authors)

Reviewers' Comments:

Reviewer #1:

Remarks to the Author:

All of the responses to my questions and comments are great. I have no further comments. I think it is a really nice result! Because APT is a new method, and the 3D nature of the data is not always clear, I wonder if putting a movie in the supplementary online section would be beneficial.